# A Review of Planned, Ongoing Clinical Studies and Recent Development of BNCT in Mainland of China

**DOI:** 10.3390/cancers15164060

**Published:** 2023-08-11

**Authors:** Zizhu Zhang, Yizheng Chong, Yuanhao Liu, Jianji Pan, Cheng Huang, Qi Sun, Zhibo Liu, Xiayang Zhu, Yujun Shao, Congjun Jin, Tong Liu

**Affiliations:** 1Beijing Nuclear Industry Hospital, Beijing 102413, China; 2Beijing Capture Tech Co., Ltd., Beijing 102413, China; 3Innovation Business Center, China National Nuclear Corporation Overseas Ltd., Beijing 100044, China; 4Neuboron Therapy System Ltd., Nanjing 211100, China; 5BNCT Center, Xiamen Humanity Hospital, Xiamen 361016, China; 6College of Chemistry and Molecular Engineering, Peking University, Beijing 100871, China

**Keywords:** BNCT, neutron source, boron agents, IHNI-1, NeuPex

## Abstract

**Simple Summary:**

Boron neutron capture therapy (BNCT) is an innovative radiation therapy that shows promise in treating certain types of cancer. Recent advancements in BNCT have focused on developing compact neutron source devices and more effective boron agents in mainland China. This paper highlights the successful transition of two compact neutron source devices from research to clinical trials, with plans underway for their registration as medical devices. Moreover, accelerator-based neutron source devices in construction and the development of boron agents are introduced. The challenges facing BNCT in practice in mainland China as a widely accepted and implemented cancer treatment are analyzed.

**Abstract:**

Boron neutron capture therapy (BNCT) is a promising cancer treatment modality that combines targeted boron agents and neutron irradiation to selectively destroy tumor cells. In mainland China, the clinical implementation of BNCT has made certain progress, primarily driven by the development of compact neutron source devices. The availability, ease of operation, and cost-effectiveness offered by these compact neutron sources make BNCT more accessible to cancer treatment centers. Two compact neutron sources, one being miniature reactor-based (IHNI-1) and the other one being accelerator-based (NeuPex), have entered the clinical research phase and are planned for medical device registration. Moreover, several accelerator-based neutron source devices employing different technical routes are currently under construction, further expanding the options for BNCT implementation. In addition, the development of compact neutron sources serves as an experimental platform for advancing the development of new boron agents. Several research teams are actively involved in the development of boron agents. Various types of third-generation boron agents have been tested and studied in vitro and in vivo. Compared to other radiotherapy therapies, BNCT in mainland China still faces specific challenges due to its limited clinical trial data and its technical support in a wide range of professional fields. To facilitate the widespread adoption of BNCT, it is crucial to establish relevant technical standards for neutron devices, boron agents, and treatment protocols.

## 1. Introduction

Boron neutron capture therapy (BNCT) is based on the nuclear reaction that occurs when Boron-10, which is a nonradioactive constituent of natural elemental boron, is irradiated with thermal neutrons (0.025 eV) to yield high linear energy transfer (LET) alpha particles (^4^He) and recoiling Lithium-7 (^7^Li)nuclei [1]. These two particles have high relative biological effectiveness (RBE), can produce lethal damage to cells, and limit the damage to the single-cell scale. Unlike conventional radiotherapies, such as X-ray, proton, or heavy ion therapy, which directly use the physical dose produced by the rays induced by the high-energy radiation device to focus on the tumor area, the neutrons in the BNCT process do not directly provide the treatment dose to the tumor. As targeted ^10^B agents are introduced, the dose of ^10^B (n, α) ^7^Li reaction is combined with the metabolism of ^10^B agents. This approach offers the potential for precise and selective tumor cell destruction.

At the end of the 20th century, BNCT gained international attention in the field of tumor radiotherapy, particularly with the development and clinical application of second-generation boron agents such as 4-Borono-L-Phenylalanine (BPA) and Sodium Mercapto-undecahydro-closo-dodecaborate (BSH) [1,2,3]. In mainland China, researchers also devoted their efforts to the development of BNCT and related fundamental research [4,5,6,7,8,9,10]. The neutron source is one of the basic conditions of BNCT research. A specially designed neutron source, which could be set in-hospital, was needed urgently for doctors and researchers. To solve this problem, several types of neutron sources were designed. The In-Hospital Neutron Irradiator-1 (IHNI-1) was designed and constructed based on the miniature neutron source reactor technique. A phase I/II clinical trial for treating malignant melanoma by BNCT was designed and carried out with the IHNI-1 during 2014–2015 [11,12]. The IHNI-1 provides a platform for BNCT-related research, such as neutron parameter measurement, new boron agent development, radiation biology research, etc. In recent years, significant progress has been made in the development and construction of accelerator-based BNCT (AB-BNCT) neutron source devices, and several devices with different technology routes are currently under construction. Notably, the NeuPex AB-BNCT System [13] has been established at the BNCT Center in Xiamen Humanity Hospital (XHH) and has commenced an investigator-initiated trial.

This work aims to provide an overview of these two BNCT projects in mainland China that have entered the clinical study phase. It will discuss the IHNI-1 system, including its design, capabilities, and clinical research conducted at the facility. It will also introduce the NeuPex system, the first AB-BNCT device in China, along with its design features and the ongoing clinical research at the BNCT Center in Xiamen Humanity Hospital. Furthermore, the article will briefly discuss the recent developments and challenges in compact neutron source devices and boron agents used in BNCT.

## 2. IHNI-1 System and Its Clinical Research

### 2.1. IHNI-1 System

The In-Hospital Neutron Irradiator-1 (IHNI-1) is a pool-tank-type reactor-based neutron source for BNCT. The reactor is based on a modified miniature neutron source reactor with a thermal power of 30 kW [6]. It was constructed by Beijing Capture Technology Co., Ltd. (BCTC) in Beijing, China, with the support and guidance of the China National Nuclear Corporation (CNNC). The IHNI-1 achieved full power in January 2010. Figure 1 shows the schematic illustration of the IHNI-1 system. The facility has a single-layer area of 175 m^2^, with one floor underground and two floors above ground, totaling approximately 500 m^2^ in building area [10]. 

The IHNI-1 is designed using low-enriched uranium deeply under a moderated reactor core with full natural convection heat release instead of a reactor coolant circulation system. The cooling and moderating agent is light water [4]. The facility is equipped with three horizontal neutron beams. One thermal neutron beam and one epithermal neutron beam are designed at the left and right side of the side reflector of the core, respectively. The thermal neutron beam could provide high-quality thermal neutrons for in vitro and in vivo BNCT studies as well as clinical studies of shallow tumors, such as cutaneous malignant melanoma [10]. The epithermal neutron beam is suitable for deeper tumor treatment, such as brain tumors, head and neck cancer, etc. [10]. At the tangential direction of the core, one measurement beam is branched out for prompt γ-ray neutron activation analysis (PGRNAA) [14]. The IHNI-1 also has two vertical neutron channels at the reactor vessel for neutron activation analysis [6]. It takes only 2–3 min for the reactor power to rise from 0 kW to 30 kW. Only one trained operator is needed for routine operation, and no nuclear waste would be released into the environment. If the IHNI-1 runs at 30 kW for 2.5 h a day and 4 days a week, one cage of fuel could be used for more than 20 years [9].

Characterization of the neutron beams of the IHNI-1 includes neutron spectra, neutron fluence rate and its spatial distribution, and the dose induced by undesired neutrons and γ-rays of free in-air neutron beams. The detail of the methods was described in the previous paper [15,16,17]. Table 1 and Table 2 show the measurement and calculated result of neutron fluence and gamma kerma rates of the thermal neutron beam and epithermal neutron beam, respectively.

### 2.2. Clinical Research at IHNI-1

Clinical research for treating malignant melanoma with BNCT was designed [11]. This clinical research was approved by the Medical Ethics Committee of the Third Xiangya Hospital of Central South University, and the research was conducted in accordance with the protocol as per ClinicalTrials.gov (No. NCT02759536). Patients accepted a biodistribution study four days before irradiation day to obtain tumor (T), normal skin (S), and blood (B) samples and measure ^10^B concentration to obtain the T/S, S/B, and T/B ratio factors using the inductively coupled plasma–atomic emission spectrometer (ICP-AES) method. The patient’s organ at risk (OAR) was normal skin. The prescription dose of normal skin was 16Gy-eq. BPA (produced by Syntagon AB, Sweden), and the dose was 350 mg/kg B.W. with an injection time of 90 min. ^10^B concentration in blood was measured before the treatment, and the result was obtained within 5 min; ^10^B concentration was assumed homogeneous in both normal skin and blood; compound biological effectiveness (CBE) factors for normal skin and tumor were 2.5 and 3.8, respectively; RBE factor for high LET particle was 3.2 [18,19]. The patients were treated with one or two irradiation fields for one BPA-fructose infusion between 120 min and 180 min after the beginning of the infusion [11].

The result of the biodistribution studies showed that the T/B ratio for three melanoma patients was 1.48–3.82. The T/B ratio of nodular metastasis melanoma was higher than superficial spreading melanoma. S/B ratio was in the range of 0.81–1.99. The maximum boron concentration in blood at the end of the BPA infusions of 350 mg/kg B.W. were in the range of 15.59–30.05 μg/g [12].

Three melanoma patients were treated with BNCT using the IHNI-1. According to the Acute Radiation Morbidity Scoring Criteria and Late Radiation Morbidity Scoring Scheme from the Radiation Therapy Oncology Group (RTOG) and the European Organization for Research and Treatment of Cancer (EORTC), only grade 1 and grade 2 acute radiation injury was observed. Two cases of early malignant melanoma showed complete response and survived in the 5 years of follow-up. One patient with Advanced nodular malignant melanoma only showed a partial response, but with no significant extension of life [11,20].

Recently, the IHNI-1 has been preparing for the registration of medical device production. One parallel two-center clinical trial with the Third Xiangya Hospital of Central South University and Hunan Cancer Hospital is planned for registration.

## 3. NeuPex System and Its Clinical Research

In August 2018, the BNCT Center at Xiamen Humanity Hospital (XHH), the first AB-BNCT center in China, was officially launched. The aim of the XHH BNCT Center was to establish a comprehensive international research hub incorporating clinical translation, treatment, training, and scientific research. The center is housed in a standalone four-story building (two above ground and two below ground) located at a corner of the Xiamen Humanity Hospital compound, adjacent to the main arterial Xianyue Road in Xiamen. Construction of the center began in May 2019, achieved structural completion in May 2020 despite pandemic-related challenges, and was operational by January 2021.

The XHH BNCT Center is equipped with an AB-BNCT device, NeuPex, developed by Neuboron Medical Group. The NeuPex system is designed with three treatment cabins, including two horizontal fixed beams and one vertical beam. The installation of the NeuPex system commenced at the end of 2020. In August 2021, the first epithermal neutron beam was successfully generated, and a series of preclinical animal experiments were carried out. In September 2022, the XHH Center was granted approval by the Ethics Committee of Xiamen Humanity Hospital to conduct its first investigator-initiated human clinical trial.

### 3.1. NeuPex AB-BNCT System

The NeuPex AB-BNCT System is an accelerator-based BNCT system composed of several precision subsystems that generate highly efficient and penetrative epithermal neutron beams. The NeuPex Block-I model is equipped with a tandem electrostatic accelerator produced by TAE Life Sciences, and its design prototype references the Russian BINP’s VITA [13].

The NeuPex Block-I model has been calibrated with defined parameters, including a proton energy of 2.35 MeV, a proton beam current of 10 mA, and a target power load of 23.5 kW, using a fixed lithium target. After the moderation, filtration, and convergence, epithermal neutrons are produced with an energy range of 10 to 20 keV at the BSA (Beam Shaping Assembly) exit plane. Figure 2 shows the schematic illustration of the NeuPex system. In contrast to traditional designs, the initial energy of the neutron beam produced by NeuPex is maintained at a relatively harder spectrum to pair with the subsequent extension collimator, allowing further scattering to approach an ideal energy level of approximately 5~10 keV.

The advantage depth of the NeuPex neutron beam can reach up to 11 cm; the advantage depth was evaluated in Snyder phantom based on the following conditions:

Blood concentration = 18 ppm

Brain-to-blood boron ratio = 1

Skin-to-blood boron ratio = 1

Bone-to-blood boron ratio = 0

Tumor-to-blood boron ratio = 3.5

At full power, the epithermal neutron flux (for ease of comparison, the adopted energy range is the same as NeuCure’s [21], i.e., 0.5 eV–40 keV) exceeds 1 × 10^9^ cm^−2^ s^−1^.

The energy spectrum of the NeuPex epithermal neutron beam can be altered by controlling the energy of the electrostatic accelerator and adjusting the thickness of the moderator, allowing it to adapt to different application scenarios, such as deep penetration for glioblastomas or shallow penetration for melanomas.

The patient positioning system employs a pair of six-axis precision robotic arms installed in the treatment room and the simulation positioning room. With a proprietary coordinate conversion and control system, NeuPex can enhance positioning efficiency and patient throughput. With two treatment cabins, NeuPex can perform 12 irradiations daily with about 8 h of operation.

### 3.2. NeuMANTA Treatment Planning System

NeuMANTA (Multifunctional Arithmetic for Neutron Transportation Analysis) is a proprietary treatment planning system developed by Neuboron Medical specifically for BNCT [22,23]. This comprehensive system encompasses front-end user interfaces, back-end image processing, and material conversion modules, as well as a full Monte Carlo-based dose calculation engine, COMPASS (COMpact Particle Simulation System).

By default, NeuMANTA uses a 1:1 conversion of Computerized Tomography (CT) image pixels to voxels. It then accurately constructs patient voxel models for subsequent dose calculations, utilizing either a mixture of International Commission on Radiological Protection (ICRP) and reliably sourced human material compositions, or material compositions derived from CT Hounsfield Unit (HU) values.

In terms of image importation, NeuMANTA supports the integration of CT, magnetic resonance imaging (MRI), and positron emission tomography (PET) images. It can accommodate the setting of non-uniform boron spatial distribution (based on PET), facilitating a more accurate depiction of boron dosage as well as a more precise evaluation of neutron distribution within the human body.

### 3.3. The First Investigator-Initiated Trial at XHH BNCT Center

Upon the successful completion of the largest-ever animal experiment in the history of BNCT, the Ethics Committee of Xiamen Humanity Hospital approved the initiation of the first investigator-initiated trial (IIT) (Chinese Clinical Trial Registration Number: ChiCTR2200066473), titled “A Single-center, Single-arm Clinical Trial of the Safety and Efficacy of Boron Neutron Capture Therapy (BNCT) for Advanced Refractory Malignant Tumors”.

The trial utilized the NeuPex AB-BNCT system, the NeuMANTA TPS system, and the boron drugs NBB-001 (known as BPA) and NBB-002 (known as ^18^F-BPA). The ^18^F-BPA PET scans (Chinese Clinical Trial Registration Number: ChiCTR2200063509) were performed at the Peking Union Medical College Hospital (Beijing) and the Tongji Hospital (Wuhan). NBB-002 was used to assess boron distribution and evaluate the T/N and T/B ratios. NBB-002 was applied as a companion diagnostic drug together with NBB-001 in the IIT.

The boron-based pharmaceutical was delivered at two specific dosages: 500 and 750 milligrams of BPA per kilogram of the patient’s body weight. The two BPA dosages were used to study the dose limit toxicity. For the cohort assigned the 500 mg/kg dosage, BPA was infused at a rate of 200 mg/kg/h over a span of 2 h, which was subsequently followed by a consistent rate of 100 mg/kg/h throughout the irradiation procedure. Conversely, for the cohort subjected to the 750 mg/kg dosage, BPA was administered at a rate of 200 mg/kg/h for 3 h, followed by a rate of 150 mg/kg/h during the irradiation process.

The irradiation was conducted using a proton beam injection set at 2.3 MeV and 8 mA, translating to a power loading of 18.4 kW targeted at the lithium substrate. The process employed an extension collimator featuring a 12 cm diameter at its exit aperture.

Key inclusion criteria of the trial were described in Table 3.

The inaugural patient was treated on 9 October 2022, with the final irradiation session of the trial taking place on 11 April 2023, establishing a duration of approximately six months for the entire trial. Overall, 14 patients participated in 18 irradiation sessions during the course of this trial. The patient group included seven individuals with tumors located in the head and neck region and seven individuals diagnosed with high-grade gliomas. Of these 14 patients, 11 were part of the 500 mg/kg BPA group, while the remaining 3 patients were included in the 750 mg/kg group. Upon completion of the final patient’s 3-month observation and evaluation period, the results of this trial will then be formally published.

## 4. Recent Development and the Challenges

In recent years, significant advancements have been made in neutron source devices and boron agents for BNCT. However, the widespread acceptance and implementation of BNCT as a cancer treatment modality face many challenges in practice.

### 4.1. Development of Compact Neutron Source Device

Two types of compact neutron source devices for BNCT have been developed and utilized in clinical studies in mainland China, as mentioned earlier. Each type has its own advantages. The miniature reactor-based BNCT neutron source offers mature technology, low construction and maintenance costs, and stable beam current. The accelerator-based BNCT neutron source has high public acceptance and relatively easy installation in the clinical environment.

With domestic advancements in accelerator technology, several accelerator-based neutron source devices with different technology routes are currently under construction. The BNCT device developed by the Institute of High Energy Physics (IHEP) of the Chinese Academy of Science uses a radio-frequency quadrupole (RFQ) linac accelerator. The first BNCT experimental device, D-BNCT01, has been built and installed with a proton energy of 3.5 MeV and an average flow intensity of 4.5 mA [24,25]. A fixed lithium target is used to generate neutrons. D-BNCT01 is carried out for experimental research on BNCT-related technologies. At the same time, D-BNCT02 is designed for clinical usage [25]. D-BNCT02 is currently being installed and commissioned at the Dongguan People’s Hospital in Guangdong province. The hospital’s BNCT treatment center building was completed in April 2023, spanning a total construction area of 18,000 square meters. The BNCT device developed by Lanzhou University (LU) also adopts the RFQ linac accelerator and is assembled in the Fujian Medical University Union Hospital. The equipment has completed commissioning, and animal experiments are currently underway [26]. The China Institute of Atomic Energy (CIAE) has adopted a cyclotron accelerator design with a gas pedal designed to induce an energy of 14 MeV and a proton beam current stronger than 1 mA [27]. This accelerator is intended for use at the Tai’an Central Hospital in Shandong Province, and the construction of the BNCT center building is expected to be completed in June 2024. Additionally, a team from Xi’an Jiaotong University (XJTU) reported the development of the RFQ linac accelerator-based BNCT device X-TANS [28,29,30,31]. Table 4 shows the detailed information of the accelerator-based neutron source devices mentioned above.

Moreover, in June 2022, China Biotechnology Services Holding Co., Ltd. signed an agreement with Sumitomo Heavy Industries and Stella Pharma Pharmaceuticals to import BNCT therapeutic devices and the related boron agent in Bo’ao, China. These devices and drugs are the world’s only approved and marketed BNCT therapeutic products.

In summary, several compact BNCT neutron source devices suitable for hospital assembly have been developed. With ongoing support from national policies, an increasing number of enterprises, research institutes, universities, and other industry–university research collaborations are engaged in research, development, and industrialization efforts related to BNCT equipment. The next steps will involve declaring BNCT neutron source devices as medical devices.

### 4.2. Development of Boron Agents

The development of boron agents is a crucial aspect of advancing BNCT clinical treatment and technology. BPA is widely used as a boron agent in BNCT clinical research worldwide. Several pharmaceutical companies in mainland China, such as Guangdong HEC Technology Holding Co., Ltd., Neuboron (Xiamen, China) Biomedicine Co., Ltd., Hainan Poly Pharm Co., Ltd., and Shenzhen Zhonghe Headway Bio-Sci Tech Co., Ltd., have the capacity for mass production of BPA. However, the registration process for BPA as a drug requires its evaluation for effectiveness and safety in combination with neutron sources.

Although second-generation boron agents such as BPA and BSH are currently utilized in clinical settings for BNCT and have demonstrated promising outcomes in oncology therapy, they fall short of being considered ideal boron agents. However, the advancement of compact neutron sources has opened up an experimental platform for the improvement of boron agents. Numerous research teams are actively engaged in the development of more efficient boron agents to address this need.

Zhibo Liu’s group from Peking University has established a chemical platform with imaging-guided boron delivery agents. Shi et al. developed coating-boronated porphyrins with a biocompatible poly (lactide-co-glycolide)–monomethoxy-poly(polyethylene-glycol) (PLGA–mPEG) micelle (BPN) [32]. However, its limited stability in plasma hampered its efficiency for drug delivery. To address this, a plasma-stable nanocarrier platform called carborane-loaded covalent organic polymers (BCOPs) was designed, exhibiting well-defined composition, good biocompatibility, and effective tumor accumulation [33]. Additionally, carborane-based covalent organic frameworks (B-COFs) loaded with imiquimod, featuring large specific surface areas and periodic structures, were developed for combined BNCT and immunotherapy [34]. Li et al. reported boron nitride nanoparticles (PTL@BNNPs) with high boron content and good biocompatibility. Boron nitride nanoparticles (BNNPs) were coated with a phase-transitioned lysozyme (PTL) to avoid potential systematic toxicity [35]. The further study also explored boron nanosheets (BNNSs) as a boron delivery platform for BNCT and chemotherapy. DOX@BNNSs were synthesized to achieve neutron irradiation-facilitated release of doxorubicin, enhancing the anti-tumor efficacy of BNCT [36]. Liposomes are efficient and clinically relevant delivery vehicles due to their excellent biocompatibility, stability, self-assembly capacity, and ability to carry large drugs. Li et al. reported carborane-derived liposome mimics, denoted as boronsome, for imaging-guided chemotherapy-assisted BNCT. Boronsome-suppressed tumor growth has safe, traceable tumor-targeted boron enrichment and concurrent chemoradiotherapy, which shows clinical potential [37]. Furthermore, progress has been made in the development of small-molecule theranostic agents for both boron delivery and cancer diagnosis. One notable advancement is the metabolically stable boron-derived tyrosine known as fluoroboronotyrosine (FBY), which closely resembles natural tyrosine. FBY shares a similar uptake mechanism with BPA as both rely on the L-type amino acid transporter (LAT-1). However, FBY exhibits higher stability and lower normal brain activity, with the capacity to deliver sufficient boron with PET-guided BNCT [38]. In a first-in-human study, FBY demonstrated favorable dosimetry and pharmacokinetic profiles [39]. A prospective single-center study investigated FBY’s characteristics in malignant brain tumors and revealed increased FBY metabolism in most neoplasms while remaining minimal in non-neoplastic lesions. Tumors with higher malignancy displayed greater FBY activity, highlighting the potential of FBY in BNCT for brain tumor treatment. In addition, Duan et al. engineered a ^157^Gd-porphyrin framework of gadolinium neutron capture therapy (GdNCT), triggering a robust anti-tumor immune response [40].

Genmei Xing’s group from the CAS Key Laboratory for Biomedical Effects of Nanomaterial and Nanosafety focuses on the development of multifunctional high boron content. Wang et al. reported Zirconium and mesotetra(4-carboxyphenyl)porphyrin (Zr-TCPP) metal–organic framework (MOF) nano-co-crystals were assembled and architected for precise BNCT of brain glioma [41]. Li et al. reported the design and application of exosome-coated ^10^B carbon dots for precise boron neutron capture therapy in a mouse model of glioma in situ [42].

Mi et al. reported block copolymer–boron cluster conjugate for effective boron neutron capture therapy of solid tumors [43]. Sun et al. reported a polyamide amine dendrimer, conjugated CD133 monoclonal antibodies, and encapsulating BSH, which could enhance the anti-tumor effect of BNCT [44]. The in vivo study results of the above third-generation boron agents are summarized in Table 5.

Some groups focused on nanocarriers have reported their in vitro study results. Dai et al. reported the in vitro study result of the folate receptor-mediated boron-10 containing carbon nanoparticles as potential delivery vehicles for BNCT of nonfunctional pituitary adenomas [45]. Zhang et al. reported the preparation of doxorubicin-conjugated ^10^B_4_C nanoparticles, which could be used to combine BNCT and chemotherapy [46]. Zhang et al. reported the preparation of a chitosan-lactobionic acid-thioctic acid-modified hollow mesoporous silica composite loaded with carborane (HMSN-S-S-CS-LA-TA) and its therapeutic effect on hepatocellular carcinoma [47].

It is worth noting that nanoparticulate systems for boron delivery have emerged as a crucial research aspect in mainland China, which has also become a significant research hotspot worldwide [48]. Although various types of third-generation boron agents have been tested and studied in vitro and in vivo, further substantial and convincing animal data are needed before they can enter clinical studies.

### 4.3. Challenges in Practice

To facilitate the widespread adoption of BNCT neutron source devices as medical devices in mainland China, it is crucial to establish relevant technical standards.

While there are general standards in place for medical devices and medical electrical equipment, specific standards for BNCT are needed. Although the IAEA technical report (IAEA-Tecdoc-1223) [49] and its updated version, “Advances in Boron Neutron Capture Therapy” [13], provide professional recommendations for the technical indicators of neutron sources, practical guidelines for each technical step are necessary. These guidelines help address the technical complexities and unique requirements associated with BNCT. They cover aspects such as the design and construction of neutron source devices, safety protocols, quality assurance measures, treatment planning procedures, patient positioning, and radiation monitoring during BNCT treatments. The development of comprehensive technical standards and practical guidelines promotes consistency, safety, and effectiveness in BNCT implementation. They provide a framework for healthcare professionals, researchers, and device manufacturers to ensure the reliable and standardized use of BNCT neutron source devices in clinical settings.

Regarding boron agents, the challenges associated with their development and registration are similar to those faced by neutron source devices. The effectiveness and safety of boron agents need to be evaluated in conjunction with the neutron source device, emphasizing the interdependence of both aspects. Collaboration among researchers, pharmaceutical companies, and neutron source platform providers is necessary for the development of new boron agents and their successful integration into BNCT.

## 5. Conclusions

In recent years, there have been significant developments and advancements in the field of boron neutron capture therapy (BNCT) in mainland China. Two notable projects, the IHNI-1 system and the NeuPex system, have entered the clinical study phase. Clinical research conducted for the IHNI-1 for treating malignant melanoma has shown promising results. The first IIT has been carried out at the XHH BNCT Center for advanced refractory malignant tumors. After the observation and evaluation period, the results of the trial will be officially published.

Multiple research institutes and collaborations in mainland China are actively involved in the development and construction of these compact neutron source devices. Promising progress has been made with devices utilizing RFQ linear accelerators and cyclotron accelerators, among others. Clinical installation and commissioning of these devices are underway, aiming to provide BNCT treatment in hospitals across the country.

The development of boron agents, essential for effective BNCT, is also advancing. BPA would be in mass production according to clinical requirements. Various research teams in mainland China are actively engaged in developing third-generation boron agents, utilizing different approaches such as boron nitride nanoparticles, boron nanosheets, liposomes, and small-molecule theranostic agents.

In summary, recent advancements in neutron source devices and boron agents for BNCT show promising progress in mainland China. However, overcoming the challenges related to technical standards and practical guidelines and promoting the development and registration of neutron sources and boron agents is crucial for the widespread acceptance and implementation of BNCT as a viable cancer treatment modality. Continued research, collaboration, and innovation are necessary to further enhance the effectiveness and accessibility of BNCT in clinical applications.

## Figures and Tables

**Figure 1 cancers-15-04060-f001:**
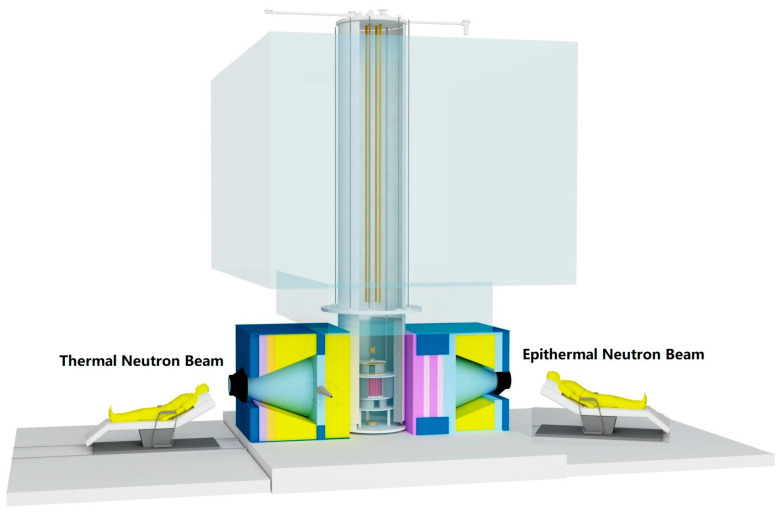
The schematic illustration of IHNI-1 system.

**Figure 2 cancers-15-04060-f002:**
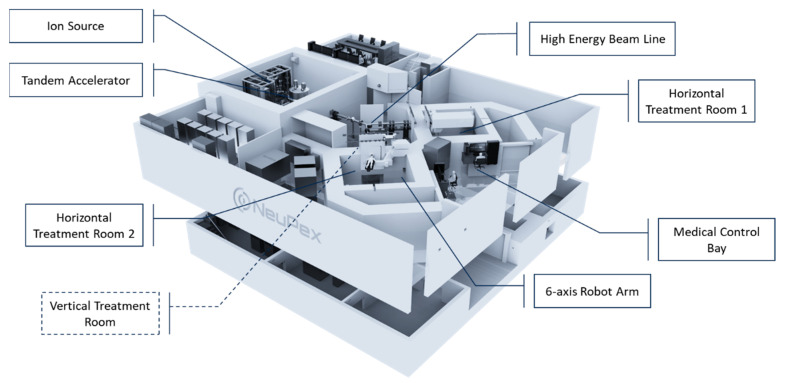
A schematic illustration of NeuPex system.

**Table 1 cancers-15-04060-t001:** The measured and calculated values of neutron fluence of thermal neutron beam and epithermal neutron beam.

	Measured Values			Calculated Values		
	*φ*_th_/cm^−2^·s^−1^	*φ*_epi_/cm^−2^·s^−1^	*φ*_f_/cm^−2^·s^−1^	*φ*_th_/cm^−2^·s^−1^	*φ*_epi_/cm^−2^·s^−1^	*φ*_f_/cm^−2^·s^−1^
Thermal beam	1.90 × 10^9^ (1 ± 0.016)	1.05 × 10^8^ (1 ± 0.027)	2.29 × 10^7^ (1 ± 0.146)	1.87 × 10^9^ (1 ± 0.018)	9.59 × 10^7^ (1 ± 0.084)	2.04 × 10^7^ (1 ± 0.151)
Epithermal beam	1.91 × 10^7^ (1 ± 0.016)	4.90 × 10^8^ (1 ± 0.027)	6.94 × 10^7^ (1 ± 0.073)	1.82 × 10^7^ (1 ± 0.121)	5.03 × 10^8^ (1 ± 0.028)	6.02 × 10^7^ (1 ± 0.083)

**Table 2 cancers-15-04060-t002:** The measured and calculated values of gamma kerma rates of thermal neutron beam and epithermal neutron beam.

	Measured Values	Calculated Values
	*K*_γ_/Gy·h^−1^	*K*_γ_/Gy·h^−1^
Thermal beam	1.01 (1 ± 0.112)	0.50 (1 ± 0.091)
Epithermal beam	1.05 (1 ± 0.112)	0.36 (1 ± 0.110)

**Table 3 cancers-15-04060-t003:** Key inclusion criteria of the first investigator-initiated trial at XHH BNCT Center.

1. All patients must meet the following requirements: A. Age between 18 and 80 years, with no gender restrictions. B. An Eastern Cooperative Oncology Group (ECOG) performance score between 0 and 2. C. The tumor-to-normal tissue boron concentration ratio (TNR) should be N > 2.5, as suggested by the L-^18^F-BPA-PET/CT examination. D. Expected survival time not less than 3 months.
2. Patients with recurrent refractory head and neck malignant tumors must also meet the following requirements: A. Histologically confirmed head and neck malignant tumors, including nasopharynx, nasal cavity, paranasal sinus, oropharynx, oral cavity, hypopharynx, and larynx. B. Being at stages III and IV and having failed standard treatment, being unwilling to accept standard treatment, or being unable to be treated in other ways, as per the guidelines of the Chinese Society of Clinical Oncology (CSCO). C. According to RECIST1.1, at least one assessable tumor lesion.
3. Patients with primary malignant brain tumors must also meet the following requirements: A. Histologically confirmed primary malignant brain tumors, CNS WHO Grade III or Grade IV. B. Patients with measurable lesions according to the RANO (2010) standards.

**Table 4 cancers-15-04060-t004:** Characteristics of accelerator-based neutron source devices developed domestically.

Developer	Name of the Project	Location	Accelerator Type	Target	Proton Beam Energy (MeV)	Beam Curren (mA)	Current Status	Citation
Neuboron	NeuPex	Xiamen Humanity Hospital, Xiamen, China	Electrostatic Tandem	Li	2.35	10	Clinical research	[13]
IHEP	D-BNCT01	Dongguan Neutron Science Center, Dongguan, China	RFQ	Li	3.5	4.5	BNCT research	[24,25]
IHEP	D-BNCT02	Dongguan People’s Hospital, Dongguan, China	RFQ	Li	2.8	20	Commissioning	[25]
LU	-	Fujian Medical University Union Hospital, Putian, China	RFQ	Li	2.5	30	Commission complete	[26]
CIAE	CIAE-14	Tai’an Central Hospital, Tai’an, China	Cyclotron	Be	14	1	Under construction	[27]
XJTU	X-TANS	-	RFQ	Li	2.5	10	Under development	[28,29,30,31]

**Table 5 cancers-15-04060-t005:** The in vivo study results of the third-generation boron agents.

Agent	Type	Injection	Max Boron in Tumor (ppm)	Tumor/Blood Ratio	Efficacy	Safety	Drug Loading Property	Imaging Property	Citation
BPN (boronated porphyrin nanocomplex)	Nanoparticle (micelle)	Intravenous injection	125.17 ± 13.54 (five-time injection)	33.85 ± 5.73 (five injections)	Tumor growth was suppressed and survival was prolonged	No obvious weight changes or abnormity of major organs	N	Y (PET-CT)	[32]
BCOPs (carborane-loaded covalent organic polymers)	Nanoparticle (micelle)	Intravenous injection	84.93 ± 2.68 (three injections)	7.46 ± 0.66 (three injections)	Tumor growth was suppressed and survival was prolonged	No significant pathological damage or changes in major organs	N	Y (PET-CT)	[33]
B-COFs (carborane-based covalent organic frameworks)	Nanoparticle (COF)	Intratumoral injection	19.8 (24 h)	N.A.	Tumor growth was delayed over 40 days and survival was prolonged	No obvious loss of body weight and the results of serum biochemical test, routine blood analysis, and Hematoxylin-Eosin staining of major organs showed negligible systemic toxicity	Y	Y (PET-CT)	[34]
PTL@BNNPs (phase-transitioned lysozyme decorated boron nitride nanoparticles)	Nanoparticle (nanosheet)	Intravenous injection	Above 120 (24 h)	2.71 ± 0.96 (24 h)	Tumor growth was suppressed and survival was prolonged	No obvious histological abnormality	N	Y (PET-CT)	[35]
BNNSs (boron nitride nanosheets)	Nanoparticle (boron nanosheet)	Intravenous injection	Above 20 (24 h)	2.4 (24 h)	Tumor growth was suppressed and survival was prolonged	No abnormalities were observed in major organs	Y	Y (PET-CT)	[36]
Boronsome (carborane-derived liposome)	Nanoparticle (liposome)	Intravenous injection	93 (12 h)	4.2 (12 h)	The tumor volume shrank to 1/5 after about 3 weeks (PARPi-boronsome) and survival was prolonged	No systemic toxicity or side effects	Y	Y (PET-CT)	[37]
FBY (boron-derived tyrosine)	Small molecular	Intravenous injection	19.59 ± 0.47 (1 h)	3.13 ± 0.50 (1 h)	Tumor growth was suppressed and survival was prolonged	No obvious weight changes or abnormity of major organs	N	Y (PET-CT)	[38]
^157^Gd-TCPP MOFs (MOF fabricated by Gd^3^+ and porphyrin derivative)	Nanoparticle (MOF)	Intratumoral injection	95.1 ± 43.7 (8 d)	Above 800 (8 d)	Tumor growth was suppressed and survival was prolonged	No obvious weight changes or abnormity of major organs	N	Y (PET-CT)	[40]
Zr-TCPP MOFs nano-co-crystals loaded with boric acids	Nanoparticle (MNC and MOFs nano-co-crystal)	Intravenous injection	67.50 ± 4.20 (2 h)	3.80 ± 0.35 (2 h)	Tumor growth was suppressed and survival was prolonged	No obvious weight changes	N	Y (PET-CT)	[41]
BCD-Exos (exosome-coated ^10^B carbon dots)	Nanoparticle (carbon dots)	Intravenous injection	107.07 ± 1.58 (4 h)	5.28 ± 0.29 (4 h)	Tumor growth was suppressed and survival was prolonged	No obvious abnormity of major organs	N	Y (Fluorescence imaging)	[42]
PEG-*b*-P(Glu-SS-BSH) (PEGylated BSH-polymer conjugate)	Nanoparticle (micelle)	Intravenous injection	128 (24 h)	~10 (24 h)	Tumor growth was suppressed	No visible damage of surrounding skin/tissue or body weight loss	N	N	[43]
PD-CD133/BSH (folate receptor-mediated boron-10 containing carbon nanoparticles)	Antibody (boron conjugated antibody)	Intratumoral injection	25.7 ± 5.8 (12 h, with 100 mg/kg BSH)	~2.7 (12 h, with 100 mg/kg BSH)	The survival of mouse was prolonged	N.A.	N	N	[44]

Abbreviations: N: no; Y: yes; N.A.: not available; PET-CT: positron emission tomography–computed tomography; d: day; and h: hour.

## Data Availability

The data presented in this manuscript have all been published and can be retrieved by going to the references indicated. There are no materials relating to this review.

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
