# Peer review of "A Review of Planned, Ongoing Clinical Studies and Recent Development of BNCT in Mainland of China"

_cancers, 2023, doi:10.3390/cancers15164060_

Round 1
Reviewer 1 Report
This content provides an overview of the first accelerator-based BNCT facility initiated in China. BNCT is a type of radiotherapy in which Japan is ahead of other countries, but the facility described in the manuscript is a very interesting introduction to the first facility outside Japan to perform BNCT. The content is not a research paper, but it is an appropriate review of the facility. However, some pictures or diagrams of an overview of the facility would be a better review. I think there are many people who would like to know more about this facility. Therefore, I think there is value in publishing this content.
I think this manuscript is written in proper English.
Author Response
We were deeply honored to have both the editor and the reviewer dedicate their time and effort to reviewing this manuscript. The valuable insights and suggestions have significantly contributed to the improvement of our study. Taking into account the mentioned points, we have diligently clarified and revised the manuscript accordingly. With these revisions, we confidently resubmit this updated version for consideration for publication in Cancers.
Sincerely,
Zizhu Zhang
Beijing Nuclear Industry Hospital, Beijing 102413, China
E-mail:zhangzizhu@bctc.cn
Comments and Suggestions for Authors
This content provides an overview of the first accelerator-based BNCT facility initiated in China. BNCT is a type of radiotherapy in which Japan is ahead of other countries, but the facility described in the manuscript is a very interesting introduction to the first facility outside Japan to perform BNCT. The content is not a research paper, but it is an appropriate review of the facility. However, some pictures or diagrams of an overview of the facility would be a better review. I think there are many people who would like to know more about this facility. Therefore, I think there is value in publishing this content.
Response
Thank you for your comment. The researchers in Japan have indeed made significant progress in the field of BNCT and have been at the forefront of BNCT research for a long time, including clinical research, BNCT neutron source devices, and complementary technologies.
The manuscript is focused on recently developing compact neutron source devices and more effective boron agents in mainland China. Pictures of the facilities and one diagram of new boron agents about the development in BNCT are added to the manuscript.
Reviewer 2 Report
The article deals with the advances in BNCT in China. It presents an overview of the impressive amount of research and the clinical trials performed and undergoing. The subject is of extreme relevance and will be of great interest for the audience interested in BNCT. Nevertheless, in my opinion, there are some issues that should be addressed before publication. Although the focus is on develpments in China, I believe it would be valuable to integrate this review in the frame of the contributions from other groups/countries, specially those that have already performed clinical trials/treatments.
General comment: The authors claim to discuss the challenges to make BNCT widely accepted. I believe that, since the article is focused in China’s developments and does not cite nor discuss with other contributions, this claim should be deleted/reformulated. With that aim, the last sentence of the simple summary, the abstract, section 4.3 and conclusions should be revised taking into account this comment.
Specific comments:
Abstract
- What do the authors refer by “low technical maturity”?
1. Introduction
- line 57: Not only “Researchers in China” are devoted to these issues. If you want to stress the contributions of chinese researchers, please rephrase.
- line 60: “A specially designed neutron source, which could be set in hospital, was needed urgently for doctors and researchers. To solve this problem, In-Hospital Neutron Irradiator-1(IHNI-1) was designed and constructed based on the miniature neutron source reactor technique.” The authors should notice that several compact neutron source have been developed and commercialized. So “to solve this problem” there has been several develpments apart from the stated in the article.
- line 67: what do you mean by “rapid development”?
- line 68: Are there no citations for the NeuPex system?
2. IHNI-1
- Line 128-130: On what base where the reported RBE and CBE factors chosen? A citation is needed.
- Why only 3 patients were enrolled in the trial?
3. Neupex
- Section 3.1
- Line 169-170: A citation is needed for the TAE-BINPS accelerator.
- Line 175: What do you mean by “traditional designs”? All the compact neutron sources are quite novel to be considered traditional. Besides, citation is needed.
- I suggest adding tables similar to those reported in section 2 (table 1 and 2).
- A figure comparing the neutron spectra for the Neupex and the IHNI-1 could be useful
- Lines 179-181: could you re-phrase to make it clearer?
- Line 183: Please include a citation for the Neucure system mentioned.
Section 3.2
There is no many detail on the TPS, so references to other papers describing the NeuManta system and its distinctive characteristics could be useful.
Section 3.3
- The inclusion criteria for the brain and H&N patients should be described.
- Line 215: was the NBB-002 used to assess boron distribution and to calculate T/N and T/B? Were the treatments performed with NBB-001?
- Line 220: Which is the rationale for choosing the two boron dosages?
Section 4.1
- Line 248: I do not agree with the following statement: “There is an urgent need for compact neutron source devices specifically designed for BNCT.”. It should be noted that there are many developments specifically designed for BNCT. If I didn’t get the author’s statement, please elaborate to make it clearer.
- I think that a table summarizing the most relevant characteristic of each development could be useful to make the section clearer and to enable comparisons. Some of the columns could be: Centre, developer, type, target, Beam energy, beam current, Current status, etc.
Section 4.2
This section is quite difficult to follow, since several developments are mentioned with short detail. I suggest elaborating a table with (for example): name of the development, type of agent (nanoparticles, small boron compounds, etc), advantages (that makes it novel in comparison with developments from other groups) and disadvantages, citation.
Section 4.3
- Line 357: I don’t understand the following statement: “BNCT faces specific challenges due to its lower technical maturity, limited clinical trial data, and wide range of technical fields”. Actually, the international consent is that, to make BNCT a widely applied technique, acceptance by clinicians is challenging and dose reporting should be standardized. Which is the situation in China in terms of BNCT adoption by oncologist and radiotherapists?
- Line 367: please include the citation to the IAEA publication, that is already published and available. Moreover, could you please elaborate why practical guidelines should be written? The IAEA publication is already quite comprehensive and contains much of this information.
Conclusions
I found the Conclusions too repetitive with what is already stated in the article, with lack of added value. I suggest re-structuring and to make an effort to make an overall conclusion comparing the two reported experiences (although they are focused in different malignancies), a general conclusion of the different developments both in terms of boron compounds and neutron sources, room for improvement and expected outcomes of the ongoing projects.
Typos, spelling comments and grammar:
- abstract: in-vivo and in-vitro should appear in italics.
- I would suggest changing “touch upon” to “briefly discuss”.
- The citation style differs in some references. For example, citation [4] (line 88) appears with other style. Also citation [1-3] (line 57).
- Please check the space between the text and the parenthesis. For example, in line 117: “was designed[11].” to “was designed [11].”.
- Table I: delete “This is a table”.
- please check that the acronyms appear defined only once. In the conclusions, there are many repetitions (e.g. BNCT in line 382).
- CT, MRI and PET are not defined. I’m not sure if the journal needs this kind of definitions.
I have already included in my report those phrases that could be improved.
Reviewer 3 Report
The paper entitled “A Brief Review of Planned, Ongoing Clinical Studies and Recent Development of BNCT in Mainland of China” by Zhang et al. is an interesting review, well-organized and well-written.
I only have few suggestions for the Authors:
1) In Table 1 caption, “The measurement and calculated values…” should be replaced with “The measured and calculated values…”. The same in Table 2 caption.
2) I suggest the Authors to add a schematic representation of the two neutron sources (IHNI-1 and NeuPex) described, to ease the reader understanding the mechanism of functioning of the systems.
3) Starting from line 304, the Authors mention several works describing the development of different nanocarriers to vehiculate boron agents. I suggest the Authors to add a line to introduce this important topic, the topic related to the growing interest towards the development of nanoparticulate systems for boron delivery, aiming at selectively carrying the boron atoms to cancer, avoiding damage at the healthy tissues, as underlined in this very recent review article: Ailuno G. et al. Boron Vehiculating Nanosystems for Neutron Capture Therapy in Cancer Treatment. Cells 2022, 11, 4029. https://doi.org/10.3390/cells11244029 I suggest the Authors to cite this paper.
4) On line 357, I suggest the Authors to replace “radiotherapy therapies” with “radiotherapies”.
Author Response
We were deeply honored to have both the editor and the reviewer dedicate their time and effort to reviewing this manuscript. The valuable insights and suggestions have significantly contributed to the improvement of our study. Taking into account the mentioned points, we have diligently clarified and revised the manuscript accordingly. With these revisions, we confidently resubmit this updated version for consideration for publication in Cancers.
Sincerely,
Zizhu Zhang
Beijing Nuclear Industry Hospital, Beijing 102413, China
E-mail:zhangzizhu@bctc.cn
Comments and Suggestions for Authors
The paper entitled “A Brief Review of Planned, Ongoing Clinical Studies and Recent Development of BNCT in Mainland of China” by Zhang et al. is an interesting review, well-organized and well-written.
I only have few suggestions for the Authors:
1) In Table 1 caption, “The measurement and calculated values…” should be replaced with “The measured and calculated values…”. The same in Table 2 caption.
Response: Thank you for the suggestion. The cation of Table 1 and Table 2 have been revised.
2) I suggest the Authors to add a schematic representation of the two neutron sources (IHNI-1 and NeuPex) described, to ease the reader understanding the mechanism of functioning of the systems.
Response: Thank you for the suggestion. Two figures have been added to show the stuction of the two neutron sources
3) Starting from line 304, the Authors mention several works describing the development of different nanocarriers to vehiculate boron agents. I suggest the Authors to add a line to introduce this important topic, the topic related to the growing interest towards the development of nanoparticulate systems for boron delivery, aiming at selectively carrying the boron atoms to cancer, avoiding damage at the healthy tissues, as underlined in this very recent review article: Ailuno G. et al. Boron Vehiculating Nanosystems for Neutron Capture Therapy in Cancer Treatment. Cells 2022, 11, 4029. https://doi.org/10.3390/cells11244029 I suggest the Authors to cite this paper.
Response: Thank you for the suggestion. Nanoparticulate systems are become ideal boron delivery for BNCT. In the last paragraph of section “4.2 Development of boron agents”,, the following comments are added: “It's worth noting that nanoparticulate systems for boron delivery have emerged as a crucial research aspect in mainland China, which have also become a significant research hotspot worldwide.” The review article of Ailuno G. et al. has been cited.
Round 2
Reviewer 2 Report
The authors have provided a point by point response and corrected the manuscript according to the suggestions. I believe the article is now suitable for publication. Please find minor comments, that are not mandatory:
1) Regarding the information of the Neupex that are still under optimization, I recommend to state it in the article, emphasising that further detail will be provided in a future work.
2) Regarding these two comments:
- Line 215: was the NBB-002 used to assess boron distribution and to calculate T/N and T/B? Were the treatments performed with NBB-001?
Response: Yes, NBB-002 (a.k.a. 18F-BPA) was used to assess boron distribution and evaluate the T/N and T/B ratio. NBB-002 was applied as a companion diagnostic drug together with NBB-001 in the IIT.
- Line 220: Which is the rationale for choosing the two boron dosages?
Response: The two BPA dosages were used to study the dose limit toxicity.
It would be useful to include these statements in the article.
Spelling suggestions:
- line 287: detail information should be changed to detailed information
- line 359: I believe there is some mixup between in-vitro and in-vivo in the text and in caption of Table 5.
- line 405: “the results of the trial are officially published”. If I understand correctly, they haven´t been published yet, so the correct phrase would be “the results of the trial will be officially published.”
Author Response
We feel great thanks for your professional review work on our article in the second round. As you are concerned, there are still several problems that need to be addressed based on the first round revised manuscript. According to your suggestions, we have made extensive corrections to our previous draft, the detailed corrections are listed below.
Sincerely,
Zizhu Zhang
Beijing Nuclear Industry Hospital, Beijing 102413, China
E-mail:zhangzizhu@bctc.cn
Comments and Suggestions for Authors
The authors have provided a point by point response and corrected the manuscript according to the suggestions. I believe the article is now suitable for publication. Please find minor comments, that are not mandatory:
1) Regarding the information of the Neupex that are still under optimization, I recommend to state it in the article, emphasising that further detail will be provided in a future work.
2) Regarding these two comments:
- Line 215: was the NBB-002 used to assess boron distribution and to calculate T/N and T/B? Were the treatments performed with NBB-001?
Response: Yes, NBB-002 (a.k.a. 18F-BPA) was used to assess boron distribution and evaluate the T/N and T/B ratio. NBB-002 was applied as a companion diagnostic drug together with NBB-001 in the IIT.
- Line 220: Which is the rationale for choosing the two boron dosages?
Response: The two BPA dosages were used to study the dose limit toxicity.
It would be useful to include these statements in the article.
Response: Thank you for the suggestion. The expression has been added in lines 232-234 and lines 236-237.
Comments on the Quality of English Language
Spelling suggestions:
- line 287: detail information should be changed to detailed information
Response: Thank you for the suggestion. The expression has been revised to “detailed information”.
- line 359: I believe there is some mixup between in-vitro and in-vivo in the text and in caption of Table 5.
Response: The caption of Table 5 has been revised as “The in vivo study results of the third-generation boron agents”.
- line 405: “the results of the trial are officially published”. If I understand correctly, they haven´t been published yet, so the correct phrase would be “the results of the trial will be officially published.”
Response: Yes, the results of the trial haven´t been published yet. The expression has been revised to “the results of the trial will be officially published” in lines 407-408.